# Early Diagnosis and Prognostic Value of Acute Kidney Injury in Critically Ill Patients

**DOI:** 10.3390/medicina55080506

**Published:** 2019-08-20

**Authors:** Diana Dobilienė, Jūratė Masalskienė, Šarūnas Rudaitis, Astra Vitkauskienė, Jurgita Pečiulytė, Rimantas Kėvalas

**Affiliations:** 1Department of Children Diseases, Medical Academy, Lithuanian University of Health Sciences, LT-44307 Kaunas, Lithuania; 2Department of Laboratory Medicine, Medical Academy, Lithuanian University of Health Sciences, LT-44307 Kaunas, Lithuania

**Keywords:** critically ill children, acute kidney injury, biomarker, uNGAL, uIL-18

## Abstract

*Background and objectives:* In hospitalized children, acute kidney injury (AKI) remains to be a frequent and serious condition, associated with increased patient mortality and morbidity. Identifying early biomarkers of AKI and patient groups at the risk of developing AKI is of crucial importance in current clinical practice. Specific human protein urinary neutrophil gelatinase-associated lipocalin (uNGAL) and interleukin 18 (uIL-18) levels have been reported to peak specifically at the early stages of AKI before a rise in serum creatinine (sCr). Therefore, the aim of our study was to determine changes in uNGAL and uIL-18 levels among critically ill children and to identify the patient groups at the highest risk of developing AKI. *Materials and methods:* This single-center prospective observational study included 107 critically ill children aged from 1 month to 18 years, who were treated in the Pediatric Intensive Care Unit (PICU) of Lithuanian University of Health Sciences Hospital Kauno Klinikos from 1 December 2013, to 30 November 2016. The patients were divided into two groups: those who did not develop AKI (Group 1) and those who developed AKI (Group 2). *Results:* A total of 68 (63.6%) boys and 39 (36.4%) girls were enrolled in the study. The mean age of the patients was 101.30 ± 75.90 months. The mean length of stay in PICU and hospital was 7.91 ± 11.07 and 31.29 ± 39.09 days, respectively. A total of 32 (29.9%) children developed AKI. Of them, 29 (90.6%) cases of AKI were documented within the first three days from admission to hospital. In all cases, AKI was caused by diseases of non-renal origin. There was a significant association between the uNGAL level and AKI between Groups 1 and 2 both on day 1 (*p* = 0.04) and day 3 (*p* = 0.018). Differences in uNGAL normalized to creatinine in the urine (uCr) (uNGAL/uCr) between the groups on days 1 and 3 were also statistically significant (*p* = 0.007 and *p* = 0.015, respectively). uNGAL was found to be a good prognostic marker. No significant associations between uIL-18 or Uil-18/uCr and development of AKI were found. However, the uIL-18 level of >69.24 pg/mL during the first 24 h was associated with an eightfold greater risk of AKI progression (OR = 8.33, 95% CI = 1.39–49.87, *p* = 0.023). The AUC for uIL-18 was 73.4% with a sensitivity of 62.59% and a specificity of 83.3%. Age of <20 months, Pediatric Index of Mortality 2 (PIM2) score of >2.5% on admission to the PICU, multiple organ dysfunction syndrome with dysfunction of three and more organ systems, PICU length of stay more than three days, and length of mechanical ventilation of >five days were associated with a greater risk of developing AKI. *Conclusions:* Significant risk factors for AKI were age of <20 months, PIM2 score of >2.5% on admission to the PICU, multiple organ dysfunction syndrome with dysfunction of 3 and more organ systems, PICU length of stay of more than three days, and length of mechanical ventilation of > five days. uNGAL was identified as a good prognostic marker of AKI. On admission to PICU, uNGAL should be measured within the first three days in patients at the risk of developing AKI. The uIL-18 level on the first day was found to be as a biomarker predicting the progression of AKI.

## 1. Introduction 

Previous studies have used widely disparate definitions for acute kidney injury (AKI) and there was a lack of uniform diagnostic criteria for AKI. This led to reporting a considerably varying incidence of AKI among children treated in a pediatric intensive care unit (PICU), ranging 4.5% to 82% [1,2,3]. Recently, the classification system by the Kidney Disease: Improving Global Outcomes (KDIGO) Workgroup created in 2012, which harmonized pediatric Risk, Injury, Failure, Loss, End Stage Renal Disease (pRIFLE) and Acute Kidney Injury Network (AKIN) criteria, is considered as the most suitable approach to diagnose AKI in a pediatric population [4]. By using unified criteria for AKI diagnosis, more accurate data on the incidence of AKI are reported, with incidence ranging from 37% to 51% [5]. Srinivasa et al. showed that the AKIN criteria were more sensitive than the pRIFLE criteria in diagnosing AKI [6]. Both criteria were found to be good in predicting PICU mortality [6]. An observational study by Sutherland et al. analyzing the incidence of AKI among hospitalized children and comparing it by using the pRIFLE, AKIN, and KDIGO criteria reported that inter-definition agreement was as low as 77% [7].

AKI remains a frequent and complex issue among hospitalized children, associated with greater patient morbidity and mortality [5,8]. Studies have reported AKI as an independent risk factor leading to increased mortality rates [9,10]. Plötz et al., found that PICU-treated children who developed AKI died five times more frequently than those who did not develop AKI [11]. Renal replacement therapy remains the treatment of choice, leading to increased treatment costs. Early recognition of renal damage, application of preventive measures, and initiation of adequate treatment are of crucial importance in order to avoid consequent progression of renal damage [12]. Modern medicine necessitates the identification of novel early AKI biomarkers that would correlate with renal cell damage and could be detected earlier than a rise in serum Cr (sCr) [12,13,14,15]. Novel AKI biomarkers should be noninvasive, easy and quickly detected, specific and sensitive for the identification of AKI, and influencing treatment strategy and predictive of disease outcomes [12,16,17]. Specific human protein urinary neutrophil gelatinase-associated lipocalin (uNGAL) and interleukin 18 (uIL-18) are one of the early biomarkers of renal damage. The levels of these biomarkers increase specifically and sensitively in early stages of AKI before a rise in the sCr level [12,13,18]. A rise of these biomarkers in urine, not serum, predicts renal damage more accurately [1,13,19,20].

During the last 10–15 years, researchers aim at identification of patient groups at different risk for developing AKI and they could undergo testing of early AKI biomarkers. In case the increased levels of these biomarkers are recorded, the application of preventive measures for AKI development or progression should be initiated [5]. Therefore, the aim of our study was to determine changes in uNGAL and uIL-18 levels among critically ill children and to identify the patient groups at the highest risk of developing AKI.

## 2. Materials and Methods

This prospective study enrolled critically ill patients aged 1 month to 18 years, who were treated at the Pediatric Intensive Care Unit (PICU), Hospital of the Lithuanian University of Health Sciences Kauno Klinikos (LSMUL KK), from 1 December 2013, to 30 November 2016. Children were included in the study if they had burns greater than 10% total body surface area with shock, underwent complex surgical procedures, sustained polytrauma, blood loss; became oliguric (urine output <1 mL/kg/h) despite fluid resuscitation within 6 h; had signs of systemic inflammatory response syndrome (SIRS) and microcirculatory dysfunction; were suspected of severe sepsis (except urosepsis); experienced clinical death and were successfully revived. Both parents signed written consent for their child to participate in the study.

The exclusion criteria were as follows: length of stay > two days in other hospital before admission to LSMUL KK, signs of renal insufficiency or anuria on admission, kidney transplantation, malignant diseases, diabetes mellitus, systemic vasculitis, rheumatoid arthritis or other systemic disease, urinary tract infection, usage of intravenous contrast material.

Illness severity on hospital admission in critically ill patients who were recruited following the inclusion criteria was assessed by the pediatric index of mortality 2 (PIM2) score. Within the first five days after admission, the amounts of fluid administered and fluid eliminated were measured. Fluid overload (FO%) on admission days 1, 3 and 5 was evaluated based on the formula:

FO% = fluid input (L) − fluid output (L) × 100%
weight (kg) on hospital admission


sCr was measured on days 1, 3, and 5 after admission as well as on discharge from hospital. NGAL, IL-18, and creatinine in urine were assessed on admission days 1 and 3.

Analysis of patient demographic (age, gender), laboratory and physical development data was performed.

Estimated glomerular filtration rate (eGFR) was calculated according to the Schwartz formula:

eGFR (mL/min/1.73 m^2^) = [height (cm) × k]/sCr (μmol/L);


Coefficient k:

k = 40 (when a child is aged ≤ 1 year), k = 49 (when a child is aged > 1 year), k = 62 (for male teenagers aged ≥ 16 years);


sCr—serum creatinine.

AKI was defined based on the classification of AKI created in 2012 by the Acute Kidney Injury Network, which unified pRIFLE and AKIN criteria [14]. An eGFR of 100 mL/min/1.73 m^2^ and 120 mL/min/1.73 m^2^ for patients younger than 1 year older than 1 year, respectively, was defined as baseline [1,21,22].

Urine samples were collected into sterile 2-mL tubes, which were frozen at −18 °C within 10–20 min and on the next day were transferred and stored at −70 °C. NGAL concentration was determined by an immunoenzymatic assay, by using commercial NGAL ELISA RD1911022000R, BioVendor, Brno, Czech Republic and IL-18 (Human IL-18 ELISA, SG-10281 SinoGeneClon Biotech Co., HangZhou, China) kits. The uNGAL level is expressed in ng/mL and normalized to uCr (uNGAL/uCr, ng/mg). The IL-18 level is expressed in ng/L normalized to uCr (uIL-18/uCr, ng/mg).

The study population was divided into two groups: those who did not develop AKI (Group 1) and those who developed AKI (Group 2).

The approval to conduct the study was issued by Kaunas Regional Biomedical research Ethics Committee (No. BE-2-15, dated 24 April 2013); on 20 June 2016, the approval to extend the study was obtained.

## 3. Statistical Analysis

Data were gathered and stored in the Excel database. SPSS version 22.0 (Armonk, NY, USA: IBM Corp.) was used for statistical analysis. The distribution of variables was assessed with the Kolmogorov–Smirnov test. The comparison of two independent groups was performed with the parametric Student *t* test or the nonparametric Mann–Whitney test. The chi-square test and Spearman exact test (small samples) were used to compare categorical variables between. The area under the curve (AUC) of the receiver operating characteristic (ROC) curve was calculated to assess the predictive ability.

Sensitivity and specificity was calculated based on these formulas:$$\begin{array}{c} \mathrm{Sensitivity}=\frac{a}{a+c}\\ \mathrm{Specificity}=\frac{d}{b+d} \end{array}$$

Here *a*—number of true positive cases; *b*—number of false positive cases; *c*—number of false negative cases; *d*—number of true negative cases.

Binary logistic regression analysis was performed to identify predictors. The level of significance was set at <0.05.

## 4. Results

The study included 107 critically ill children aged from 1 month to 18 years. There were 68 boys (63.6%) and 39 girls (36.4%). The mean age of the patients was 101.3 (SD 75.9) months.

All the patients were treated in the PICU. The mean PICU length of stay was 7.91 days (SD 11.07, range 1–66 days, median 4 days). The mean hospital length of stay was 31.29 (SD 39.09, median 18.5 days). Gender (*p* = 0.729 and *p* = 0.613) and age (*p* = 0.78 and *p* = 0.741) had no significant impact on the length of stay in PICU or hospital.

Of all the patients enrolled in the study (*n* = 107), 32 (29.9%) developed AKI. In 90.6% of the cases (29 out of 32), AKI was documented within the first three days from hospital admission.

In all cases, AKI was caused by diseases of non-renal origin. The groups were matched by diagnoses (Table 1).

The detailed analysis of both groups is shown in Table 2.

Figure 1 shows the distribution of patients’ ages by groups.

With the help of logistic regression analysis, risk factors for developing AKI were identified (Table 3).

Changes in the levels of early biomarkers uNGAL and uIL-18 on days 1 and 3 and the uNGAL and uIL-18 levels on days 1 and 3 normalized to uCr in patients without and with AKI are shown in Table 4.

There was no significant difference in the number of patients who developed multiple organ dysfunction between the groups (34 vs. 18, *p* = 0.301). Vasopressor use did not differ significantly between the groups (26 vs. 12, *p* = 0.779).

There was no significant difference in the number of patients who needed mechanical ventilation between the groups (43 vs. 23, *p* = 0.16); however, the length of mechanical ventilation in Group 2 was significantly greater (4.63 vs. 21.09 days, *p* = 0.017).

The mean FO% on days 1, 3, and 5 were not significantly different between Groups 1 and 2. There were significantly more patients in Group 2 who had FO > 15% on day 3 and FO > 10% on day 5. Logistic regression analysis showed that FO > 15% on day 3 and FO > 10% on day 5 were significantly associated with the risk of AKI development (Table 5).

Multivariate logistic regression analysis found FO > 15% on day 3, dysfunction of 3 and more organ systems, and any of the diagnoses (sepsis, burns, asystole, prolonged seizures, polytrauma, gastroenterocolitis, meningitis) to be associated with the risk of AKI development (Table 6).

uNGAL and uNGAL/uCr measured on days 1 and 3 after hospital admission was significantly associated with the development of AKI. However, there were no significant associations between these biomarkers and AKI severity, its progression, and disease outcome.

Analysis of 32 patients who developed AKI showed that the uIL-18 level of >69.24 pg/mL detected within the first day was associated with an 8-fold increased risk of AKI progression (OR = 8.33, 95% CI = 1.39–49.87, *p* = 0.023). The area under the ROC curve was 73.4%; sensitivity and specificity were 62.59% and 83.3%, respectively.

## 5. Discussion

AKI is a life-threatening clinical syndrome caused by a sudden and rapidly progressing impairment of renal function (most commonly reversible), characterized by the retention of nitrogenous waste products in blood as well as various alterations in fluid, electrolyte and acid-base homeostasis. Many authors investigating AKI causes have noticed that in 50–80% of the cases, the cause of AKI is not a renal disease, but other disease (so-called secondary AKI), and such AKI develops as a consequence of the main disease [2,5,9]. Comparison of PICU-treated children who developed AKI with those who did not develop AKI revealed no significant difference in the distribution of patients by diagnosis between the groups [23]. Our study also showed that in all cases (100%), the cause of AKI was not a renal disease and there was no significant association between diagnosis and AKI development.

Literature reports the following risk factors having an impact on AKI development: younger age, severe condition requiring treatment at the PICU, need for vasopressors or respiratory support, multiple organ dysfunction, comorbid chronic conditions, fluid overload, and use of nephrotoxic medications [5,9,24]. Our study showed that PICU length of stay > three days, bed days >55, length of mechanical ventilation > five days are risk factors significant associated with AKI development. These risk factors also show the child severity of disease. The more severe the child is ill, the more likely the lesions (hypoxic, hypovolemic, inflammatory or toxic) may be. The polyethiological mechanism of the lesion and greater illness burden are at higher AKI risk. Most probably due to the fact that maturation of the immune system is incomplete at young age, children at this age are vulnerable to generalized infections. At the age of 2 years, there is continuous development and maturation of glomerular and tubular function [5]; therefore, the kidney of a child at such age is vulnerable to hypoxic and nephrotoxic injury. The results of our study are in line with those reported by other researchers. According to our study critically ill patients who were younger than 3 years developed AKI significantly more frequently despite the fact the time from the disease onset to hospitalization was shorter. We have shown that the course of disease and occurrence of possible complications are influenced not only by patient’s age, but also by the severity of condition on hospital admission, assessed by the PIM2 score. Zwiers et al. reported that patients who developed AKI had a significantly greater PIM2 score on admission to the PICU than those without AKI (*p* = 0.011) [18]. However, the study on children with severe septic shock by Plötz et al. reported that there was no significant difference in the PIM2 scores between patients with and without AKI (4.5% vs. 4.2%, *p* > 0.05) [25]. On the other hand, the findings of this study showed that diuresis per hour was lower and cumulative fluid balance per day was greater in the AKI group than the non-AKI group [25].

Massive infusion therapy is necessary at the initial treatment stages of many critical conditions. A prospective study by Soler et al., which investigated 266 critically ill children treated in the PICU reported that FO ≥10% was directly linked to the severity of illness and mortality [26]. Li et al. found that a 1% increase in early fluid accumulation was associated with a 1.36-fold greater risk of PICU mortality [27]. As soon as AKI develops, free water and sodium excretion becomes impaired and this leads to fluid accumulation in the body. In turn, fluid accumulation results in tissue edema contributing to the progressive dysfunction of not only kidney, but also other organs [27,28]. Our study showed that FO of >15% and >10% on day 3 and on day 5, respectively, was significantly associated with AKI development.

Given that the prediction of possible AKI development is difficult, researchers pay considerable attention to the search for early AKI biomarkers that would correlate with renal cell damage [12,13,14,15]. Over the last decade, many researchers have investigated early AKI biomarkers such as NGAL and IL-18 in urine. Many studies have drawn conclusions that in patients with various diseases treated in the PICU, uNGAL is a good predictive biomarker of AKI, associated with the severity of AKI and outcome [1,10,12,29,30,31]. There are investigations that involved children only with one disease, for example those who sustained sepsis or underwent cardiac surgery. The patient population after cardiac surgeries represents a more homogenous group, when the timing and mechanism of kidney injury are known. Therefore, the results obtained in such patient cohort should not be compared with those findings documented in the critically ill, PICU-treated pediatric population when the exact timing of kidney injury is not known and the mechanism of injury can be different [16]. In the course of sepsis, the level of NGAL in serum can rise due to its increased production and excretion from neutrophils, macrophages, or epithelial cells into systemic circulation [32], and no significant association between sNGAL and development of AKI was established [33]. An increase in urine NGAL directly correlates with renal tubular damage [5,29,33]. An observational study by Di Nardo et al. has demonstrated that uNGAL is a good prognostic biomarker of AKI in children with sepsis [33]. Our study showed that uNGAL was significantly associated with AKI not only on day 1, but also after 48 h and found uNGAL to be a good predictive biomarker of AKI development. Du et al. evaluated five urinary biomarkers of AKI in 252 patients who arrived at an emergency center. It was shown that uNGAL had good accuracy to predict patients who developed AKI when sCr levels (AUC 0,7) [34]. In 2015, the study by McCaffrey et al. reported that uNGAL level directly correlated with AKI [1]. However, there are studies reporting no significant difference in the NGAL comparing patients who developed AKI and did not develop AKI [23]. This confirms the necessity to continue more studies on uNGAL of children admitted in PICU.

While analyzing pro-inflammatory cytokines, researchers have noted a considerable increase in the IL-18 level in the urine of critically ill patients. Numerous studies have investigated associations between IL-18 and severity of AKI as well as disease outcome [31,32,35]. Washburn et al. reported that an increase in the uIL-18 level after 24 and 48 h from disease onset was not only associated with an early development of AKI and its severity, but also uIL-18 rose to a level higher than control levels before a significant rise in serum Cr (*p* = 0.008) [35]. uIL-18 is not only an early marker of AKI, but also an independent predictor of mortality [35]. As uIL-18 is an inflammatory marker, the time from disease onset to uIL-18 measurement has an impact on an increase in the uIL-18 level. A prospective study on pediatric patients presenting to an emergency center found no correlation between uIL-18 and development of AKI [34]. Our study revealed no association between uIL-18 or uIL-18/uCr on days 1 or 3 and development of AKI too. However, statistically significant results were obtained in the group of patients with AKI, showing that the uIL-18 level of >69.24 pg/mL detected within the first day was associated with the progression of AKI. Our study showed that uIL-18 is a good prognostic marker of AKI progression. Our AKI group was quite small. That could cause difference in results comparing with other authors results. We think it is necessary to continue the investigation regarding uIL-18 changes in critically ill children’s group in order to get more statistically significant results.

The opinion exists that early AKI biomarkers have to be investigated not in all patients, but only in those who are at risk of acute renal damage development. Identification of patients’ groups at high risk for developing AKI is advocated [5]. The assessment of early AKI biomarkers should be performed in high-risk patients on hospital admission and later repeated after 24 or 48 h [5]. Our findings show that being aged < 20 months, PIM2 score of >2.5% on PICU admission, multiple organ dysfunction involving three and more organs, PICU bed days longer than three days, and mechanical ventilation of >5 days are associated with a high risk for developing AKI. Based on our study, we advocate to investigate uNGAL in critically ill, PICU-treated patients within the first three days and for uIL-18 assessment on the first day after hospital admission as during this period the highest incidence of AKI was documented.

An initial increase in the levels of AKI biomarkers or a consequent rise in their level with time allows suspecting AKI early. Once the results on biomarkers, suggestive of AKI, have been obtained, it is purposeful to correct a treatment strategy. The administration of nephrotoxic medications in such patients is discontinued, fluid overload is avoided, and early decision on the indications for the initiation of renal replacement therapy is taken if the sCr level rises as well [5]. Early recognition of renal damage, application of preventive measures, and initiation of adequate treatment are of crucial importance in order to avoid the development of AKI, to reduce pediatric mortality and treatment costs, and to improve disease outcomes [5,36].

## 6. Conclusions

Being aged < 20 months, a PIM2 score of >2.5%, multiple organ dysfunction involving three and more organ systems, PICU length of stay of >three days, and length of mechanical ventilation of >five days were found to be associated with an increased risk of AKI development. uNGAL was identified as a good prognostic marker of AKI. On admission to PICU, uNGAL should be measured within the first three days in patients at the risk of developing AKI. The uIL-18 level on the first day was found to be as a biomarker predicting the progression of AKI.

## Figures and Tables

**Figure 1 medicina-55-00506-f001:**
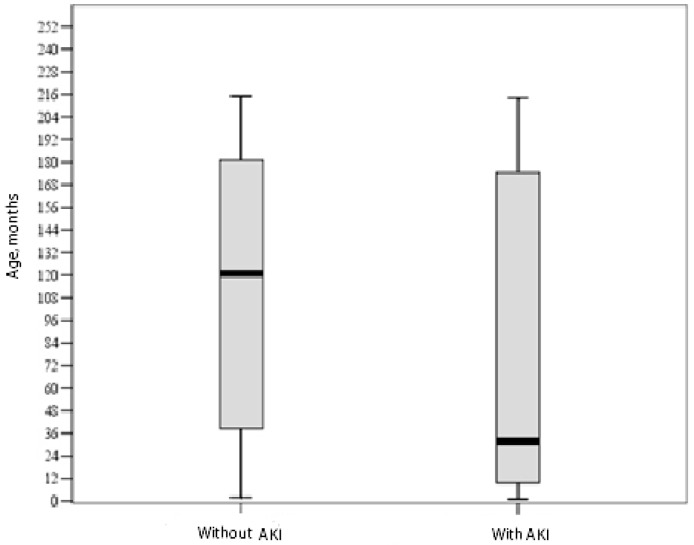
Age distribution in the groups (*p* = 0.009, nonparametric Mann–Whitney test). AKI, acute kidney injury.

**Table 1 medicina-55-00506-t001:** The distribution of critically ill patients without and with acute kidney injury (AKI) by diagnoses.

Diagnosis	Group 1 without AKI (*n* = 75, 70.1%)	Group 2 with AKI (*n* = 32, 29.9%)
Sepsis	18 (16.8)	9 (8.4)
Burns	9 (8.4)	4 (3.7)
Seizures	1 (0.9)	2 (1.9)
Intoxication	3 (2.8)	1 (0.9)
Arrhythmia	1 (0.9)	0 (0)
Complicated IVRA	2 (1.9)	0 (0)
Trauma	23 (21.5)	11 (10.3)
Blood loss	7 (6.5)	1 (0.9)
Hypovolemia	2 (1.9)	2 (1.9)
Meningitis	2 (1.9)	2 (1.9)
Other	7 (6.5)	0 (0)

Values are number (percentage). IVRA, infectio virosa respiratoria acuta.

**Table 2 medicina-55-00506-t002:** Characteristics of the study population by groups.

Characteristic	Group 1 without AKI (*n* = 75, 70.1%)	Group 2 with AKI (*n* = 32, 29.9%)	*p*
PIM2, score			
Mean ± SD	8.79 ± 18.51	12.46 ± 19.10	0.035
Median (25–75%)	2.8 (1.2–6.4)	5.7 (2.4–13.6)	
Duration of disease until hospitalization, days			
Mean ± SD	0.89 ± 1.86	0.53 ± 1.08	0.609
(range)	(0–11.0)	(0–5.0)	
PICU length of stay, days			
Mean ± SD	6.36 ± 7.66	11.47 ± 16.06	0.053
Median (25–75%)	4.0 (2.0–7.0)	6.0 (3.0–11.75)	
Bed days			
Mean ± SD	24.29 ± 23.51	48.84 ± 58.84	0.028
Median (25–75%)	17.0 (10.0–29.0)	23.0 (16.0–68.25)	

Continuous data were compared with the nonparametric Mann–Whitney test; categorical data, with Pearson chi-square test. SD, standard deviation; PICU, pediatric intensive care unit.

**Table 3 medicina-55-00506-t003:** Risk factors associated with AKI development.

Variable.	OR	95% CI	*p*
Age of <20 months	7.39	2.69–20.28	0.001
PIM2 score > 2.5	2.92	1.16–7.32	0.001
PICU length of stay >three days	2.42	0.99–59.27	0.05
Bed days > 55	5.23	1.70–16.02	0.002
Length of mechanical ventilation > 5 days	4.90	1.63–14.82	0.003
Multiple organ dysfunction (3 and more organ systems)	3.98	1.54–10.32	0.003
uNGAL/uCr of >8.22 ng/mg on day 1	3.76	1.58–8.94	0.002
uNGAL of >4.3 ng/mL on day 3	3.60	1.51–8.59	0.003

OR, odds ratio; CI, confidence interval.

**Table 4 medicina-55-00506-t004:** Distribution AKI biomarkers according to groups.

Variable	Group 1 (Patients without AKI) (*n* = 75)	Group 2 (Patients with AKI) (*n* = 32)	*p*
uNGAL on day 1			
Mean ± SD, ng/mL	5.98 ± 8.74	15.9 ± 27.03	0.04
Median (25–75%)	2.5 (0.53–6.78)	2.99 (1.44–10.45)	
uNGAL on day 3			
Mean ± SD, ng/mL	4.80 ± 6.23	10.07 ± 12.08	0.018
Median (25–75%)	1.84 (0.42–6.88)	7.56 (0.79–12.65)	
uNGAL/uCr on day 1			
Mean ± SD, ng/mg	20.35 ± 46.13	94.05 ± 200.38	0.007
Median (25–75%)	4.67 (1.1–14.11)	12.10 (2.47–90.27)	
uNGAL/uCr on day 3			
Mean ± SD, ng/mg	14.08 ± 25.25	42.06 ± 78.56	0.015
Median (25–75%)	3.94 (1.79–14.66)	12.48 (2.62–14.48)	
uIL-18 on day 1			
Mean ± SD, ng/L	62.33 ± 15.51	62.49 ± 10.15	0.959 ^a^
uIL-18 on day 3			
Mean ± SD, ng/L	61.46 ± 17.22	60.75 ± 12.95	0.834 ^a^
uIL-18/uCr on day 1			
Mean ± SD, ng/mg	0.20 ± 0.21	0.37 ± 0.43	0.062
Median (25–75%)	0.13 (0.06–0.21)	0.20(0.07–0.63)	
uIL-18/uCr on day 3			
Mean ± SD, ng/mg	0.24 ± 0.25	0.25 ± 0.27	0.509
Median (25–75%)	0.13 (0.06–0.35)	0.15 (0.08–0.34)	

*p* data were compared with nonparametric Mann–Whitney test; ^a^
*p*, with Student‘ *t* test.

**Table 5 medicina-55-00506-t005:** The influence of fluid overload on AKI development.

Variable	OR	95% CI	*p*
FO > 15% on day 3	6.76	1.24–36.93	0.024
FO > 10% on day 5	6.63	1.59–27.62	0.004

FO, fluid overload.

**Table 6 medicina-55-00506-t006:** Multivariate logistic regression analysis model.

Variable	OR	95% CI
FO > 15% on day 3	5.77	0.99–33.82
Any of diagnoses	6.82	1.13–41.05
Multiple organ dysfunction of ≥3 organ systems	20.01	3.26–122.97

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
