# Peer review of "Early Diagnosis and Prognostic Value of Acute Kidney Injury in Critically Ill Patients"

_medicina, 2019, doi:10.3390/medicina55080506_

Round 1

Reviewer 1 Report

The article is well written and presented but does not offer new findings or conclusion.

The authors have listed diagnosed associated with AKI in Table 1. All of which are etiologies for prerenal AKI. There is limited value of biomarkers in predicting AKI. Is this correct? Could this have affected the sensitivity and specificity of their findings?

What variables were controlled for multivariable logistic regression?

Author Response

Notes to reviewer 1

Point 1. The article is well written and presented but does not offer new findings or conclusion.

Response 1. We would like to thank the reviewer for spent time and very valuable comments. This prospective study has been done with critically ill children in Lithuania for first time. We have identified a risk group, among critically ill children, for whom biomarker of AKI should be investigated. This is considered to be an important finding for a targeted investigation. The following conclusions from our work are very useful for doctors practice.

Point 2. The authors have listed diagnosed associated with AKI in Table 1. All of which are etiologies for prerenal AKI. There is limited value of biomarkers in predicting AKI. Is this correct? Could this have affected the sensitivity and specificity of their findings?

Response 2: Thanks to reviewer for valuable notes. Our study was prospective, that’s why we couldn’t predict the etiology of AKI. AKI may be polyethiological in the critically ill children group and the mechanisms of AKI and the time of onset are unknown. It could influence the sensitivity and specificity of the results. In order to be precise, regarding sensitivity and specificity, another study, which would compare children with prerenal and renal AKI.

Point 3. What variables were controlled for multivariable logistic regression?

Response 3: Variables ( FO>15 ٪ on day 3, any of diagnoses, multiple organ dysfunction of ≥3 organ systems) were used in multivariable logistic regression model.

Reviewer 2 Report

In this study the author determine changes in uNGAL and uIL-18 levels among critically ill children and to identify the patient groups developing AKI. This study seemed useful for medication, however there are several point must be improved.
I will reconsider after major revision.

All figures are not clear. The author should show up clearly of figures.

The author must indicate the means of abbreviations. There are not explain what is uNGAL and uCr.

In the result section, the author did not mention to Figure 1 and Table2. If these data are not necessary, the author should remove from manuscript.

Table3: The PICU length of stay and Bed days are showed as risk factor of AKI. However, these alterations seemed cause by AKI. The author should discuss about relationship between these alteration and AKI.

Table 6: The aim of our study was to determine changes in uNGAL and uIL-18 levels among critically ill children and to identify the patient groups developing AKI. This table seemed no relation to purpose of this study.

Line 134: What is C’?

In Discussion section, it is not clear that what is important and what the author want to explain. There are many explain about citation but few comment about authors result while the main point of discussion is authors result.

Line255-256: Many researchers have investigated the NGAL and IL-18 as early AKI biomarkers. What is novelty and originality of this study?

Line 270-273: The author showed uNGAL is associated with AKI, but other researcher has reported NGAL is not associated AKI. The author must add a discussion concerning to this point.

Line 286-288: The author discussed about the factor of disagreement the result of IL-18. If the disagreement caused by these reagents differ in their sensitivity and specificity, IL-18 is not useful for biomarker because it is impossible for use same reagent in worldwide

Author Response

Notes to reviewer 2

Point 1. All figures are not clear. The author should show up clearly of figures.

Response 1. We would like to thank reviewer for very valuable comments. We agree that some figures are not so clear because we wanted to include a lot of information in them. According to your comment we changed figures 2-4 to the new table No 4 in line 165 with more clear results.

Point 2. The author must indicate the means of abbreviations. There are not explain what is uNGAL and uCr.

Response 2. Thank you for inaccuracy you found. We made changes according to reviewers comment: uCr – urine creatinine concentration (we wrote in line number 28), uNGAL - neutrophil gelatinase-associated lipocalin (we wrote in line 13).

Point 3. In the result section, the author did not mention to Figure 1 and Table2. If these data are not necessary, the author should remove from manuscript.

Response 3. Thank you for the comment. Figure 1 data shows the difference of patients age in group 1 and group 2. According to these results younger children were identified as a risk group for AKI. It is one of the conclusions. In row 232-233 we discuss about it. We think it is valuable result and we would like not to remove it from the article. We made changes in table 2. We left data which are analyzed in detail, according to which conclusions were made. Not important data were removed.

Point 4. Table3: The PICU length of stay and Bed days are showed as risk factor of AKI. However, these alterations seemed cause by AKI. The author should discuss about relationship between these alteration and AKI.

Response 4. We are grateful for the notice, that our data (PICU length of stay >3 days, bed days >55 - risk factors for AKI) were not mentioned in the discussion section. We made changes according your comment (line 226)   

Point 5. Table 6: The aim of our study was to determine changes in uNGAL and uIL-18 levels among critically ill children and to identify the patient groups developing AKI. This table seemed no relation to purpose of this study.

Response 5. Thank you for the comment. We agree with you and made changes according to your note (Table number 6 and line 208-215 were removed).

Point 6. Line 134: What is C’?

Response 6. C in the line 134 is number of false negative cases. The explanation is written in line 136-137.

Point 7. In Discussion section, it is not clear that what is important and what the author want to explain. There are many explain about citation but few comment about authors result while the main point of discussion is authors result.

Response 7. Thank you for your comment. Changes were done.

Point 8. Line 255-256: Many researchers have investigated the NGAL and IL-18 as early AKI biomarkers. What is novelty and originality of this study?

Response 8. We would like to thank the reviewer for spent time and valuable comments. This prospective study has been done with critically ill children and at the first time in Lithuania. However, we have identified a risk group, among critically ill children, for whom biomarker of AKI should be investigated. This is considered to be an important finding for a targeted investigation. The following conclusions from our work are very useful for doctors.

Point 9. Line 270-273: The author showed uNGAL is associated with AKI, but other researcher has reported NGAL is not associated AKI. The author must add a discussion concerning to this point.

Response 9. Thank you for the comment. We made changes according to the comment. We add discussion in line 271.

Point 10. Line 286-288: The author discussed about the factor of disagreement the result of IL-18. If the disagreement caused by these reagents differ in their sensitivity and specificity, IL-18 is not useful for biomarker because it is impossible for use same reagent in worldwide

Response 10. Thank you for this point. We removed mentioned sentence. Our AKI group was quite small. That could cause difference in results comparing with other authors results. We thing that it is necessary to continue the investigation regarding IL-18 changes in urine in critically ill children’s group in order to get more statistically significant results. New data and systematic review would help to standardize methodology and will help to make less mistakes in the future.

Round 2

Reviewer 2 Report

This paper is well revised and it is enough to publish.